# The Fast Discrete Interaction Approximation Concept

**Vladislav Polnikov** 

A.M. Obukhov Institute of Atmospheric Physics of RAS, Moscow 119017, Russia; polnikov@mail.ru

**Abstract:** Hasselmann and coauthors proposed the discrete interaction approximation (DIA) as the best tool replacing the nonlinear evolution term in a numerical wind–wave model. Much later, Polnikov and Farina radically improved the original DIA by means of location all the interacting four wave vectors, used in the DIA configuration, exactly at the nodes of the numerical frequency–angular grid. This provides a nearly two-times enhancement of the speed of numerical calculation for the nonlinear evolution term in a wind–wave model. For this reason, the proposed version of the DIA was called as the fast DIA (FDIA). In this paper, we demonstrate all details of the FDIA concept for several frequency–angular numerical grids of high-resolution with the aim of active implementation of the FDIA in modern versions of world-wide used wind–wave models.

**Keywords:** wind–wave modeling; nonlinear waves; kinetic integral; interacting waves; optimal configuration

## 1. Introduction

Nonlinear interactions between waves play a very important role in description of wind–wave evolution governed by the equation [1]

$$\frac{D}{Dt}N(\mathbf{k}, \mathbf{x}, t) = IN(N_{\mathbf{k}}) + NL(N_{\mathbf{k}}) - DISS(N_{\mathbf{k}}),$$

where $D/Dt$ is the total derivative operator, $N_{\mathbf{k}} \equiv N(\mathbf{k}, \mathbf{x}, t)$ is the wave-action spectrum in the wave vector $\mathbf{k}$-space, at location $\mathbf{x}$, and time $t$; $IN$, $NL$, $DISS$ are the evolution terms responsible for the input, conservative nonlinear transfer among wave components, and dissipation of wave action, respectively. The nonlinear evolution term $NL$ is described by the six-fold Hasselmann kinetic integral $I_{NL}$ with a very complicated integrand [2]

$$\frac{\partial N(\mathbf{k}_4)}{\partial t} = I_{NL}(N) \equiv$$
$$\equiv 4\pi \int M_{1,2,3,4}^2 \{N(\mathbf{k}_1)N(\mathbf{k}_2)[N(\mathbf{k}_3) + N(\mathbf{k}_4)] - N(\mathbf{k}_3)N(\mathbf{k}_4)[N(\mathbf{k}_1) + N(\mathbf{k}_2)]\}\delta_{1+2-3-4}d\mathbf{k}_1 d\mathbf{k}_2 d\mathbf{k}_3 \tag{1}$$

where $M_{1,2,3,4}^2 \equiv M^2(\mathbf{k}_0, \mathbf{k}_1, \mathbf{k}_2, \mathbf{k}_3)$ is the second power of the matrix elements corresponding to the four-wave nonlinear interactions, $\delta_{1+2-3-4} \equiv \delta(\sigma_1 + \sigma_2 - \sigma_3 - \sigma_4)\delta(\mathbf{k}_1 + \mathbf{k}_2 - \mathbf{k}_3 - \mathbf{k}_4)$ is the Dirac delta-function responsible for the resonant feature of the four-wave interactions, and $\sigma_i = \sigma(\mathbf{k}_i)$ is the radian frequency of the wave component with wave vector $\mathbf{k}_i$. Due to this complicity, the calculation of integral $I_{NL}$ takes too much time; therefore, to be used in a practical numerical wind–wave model, the integral should be replaced by some theoretically justified approximation. The best approximation was proposed by Hasselmann et al. [3], based on replacing integral $I_{NL}$ by the only configuration of four interacting waves, located at a singular subsurface. This subsurface in the six-fold $\mathbf{k}$-space is defined by the resonance conditions

$$\mathbf{k}_1 + \mathbf{k}_2 = \mathbf{k}_3 + \mathbf{k}_4, \tag{2a}$$

$$\sigma_1 + \sigma_2 = \sigma_3 + \sigma_4. \tag{2b}$$

The wave vectors $\mathbf{k}_i$ are usually represented in the frequency–angular space, $(\sigma, \theta)$, where the wave frequencies $\sigma_i$ are related to $\mathbf{k}_i$ by the dispersion relation, in the deep-water case having the kind

$$\sigma(\mathbf{k}_i) = \sigma_i = (gk_i)^{1/2}. \tag{3}$$

The proposed approximation is named as the discrete interaction approximation (DIA). Example of the four vectors configuration (which is usually called as a quadruplet) used in the original DIA [3] is schematically shown in Figure 1.

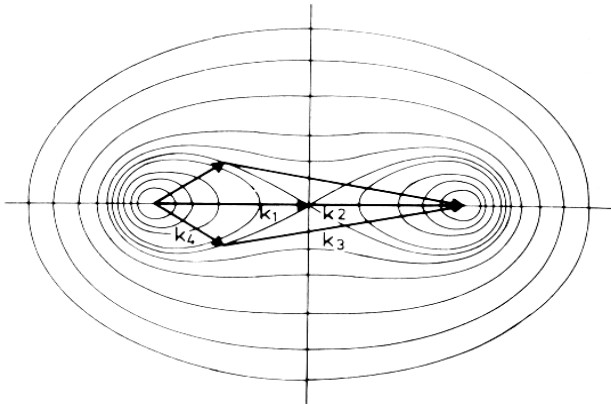

**Figure 1.** The configuration used in the original discrete interaction approximation (DIA). Two upper vectors mean the mirror image for k$_4$ and k$_3$.

It is worthwhile to mention that there are a lot of papers devoted to optimization the performance of the NL-term calculations (e.g., [4–11], see also a list of special workshops on wind wave evolution [11]). Some certain approaches to simplify the NL-term calculations, different from the DIA (except the two-scale approximation (TSA) [5,6] derived later), are presented in our paper [12], where a radical preference of the DIA was shown. This is why we dwell on the features of the DIA.

Regarding the other methods, we would say the following.

Tamura et al. [4] studied the impact of non-linear energy transfer on realistic wave fields of the Pacific Ocean using the Simplified Research Institute of Applied Mechanics (SRIAM) model, which was developed to accurately reproduce non-linear source terms with lower computational cost than more rigorous algorithms, and the widely used the DIA method. Comparison of the model with buoy observations revealed a negligible difference in significant wave heights but pronounced bias in peak frequency with the DIA. The analysis of spectral shape indicated that the SRIAM method can quantitatively capture the overshoot phenomena around the spectral peak during wave growth.

The TSA method [5] resides in shearing the whole wave spectrum under the kinetic integral in two parts: a high frequency and the energy containing ones. This allows excluding a reasonable value of the quadruplets from the NL-term calculations, saving accuracy. The TSA has recently been presented as a new method to estimate nonlinear transfer rates in wind waves, and has been tested for idealized spectral data, as well as for observed field measurements [6]. The TSA has been implemented in the wind–wave model WAVEWATCH III and shown to perform well for wave spectra from field measurements, even for cases with directional energy shearing, compared to the DIA.

Regarding the DIA, the papers by van Vledder [7] and Tolman [8,9], devoted to an extension of the DIA to multi-configuration versions, are the closest to our topic. Tolman [8] presented a generalized multiple DIA (GMD) which expands upon the DIA by (i) expanding the definition of the representative quadruplet, (ii) formulating the DIA for arbitrary water depths, (iii) providing complimentary deep and shallow water scaling terms, and (iv) allowing for multiple representative quadruplets. The GMD is rigorously derived to be an extension of the DIA, and is backward compatible with it. The free parameters of the GMD have been optimized holistically, by optimizing full model behavior in the

WAVEWATCH III wave model [9]. The results showed that in deep water, GMD configurations can be found which remove most of the errors of the DIA. Most of these improvements were implemented in a new version (4.18) of the WAVEWATCH code.

In these papers [7–9] (and further elaborations of them), they did not avoid the main shortage of the DIA—the location of all the quadruplets exactly at a singular subsurface defined by the resonant conditions (2).

The shortages of the original DIA become clear if we consider details of the original DIA.

## 2. Details of the Discrete Interaction Approximation

The quadruplet used in the original DIA in the polar coordinates $(\sigma, \theta)$ has the following parameters (see Figure 1):

(1)    $\mathbf{k}_1 = \mathbf{k}_2 = \mathbf{k}$, where wave vector $\mathbf{k} = (\sigma, \theta)$ is located at the node of the numerical grid $\{\sigma_i, \theta_j\}$;

(2)    $\mathbf{k}_3 = \mathbf{k}_+$, where $\mathbf{k}_+$ is represented by

$$\sigma_+ = \sigma(1 + \lambda) \text{ and } \theta_+ = \theta + \Delta\theta_+; \tag{4}$$

(3)    $\mathbf{k}_4 = \mathbf{k}_-$, where $\mathbf{k}_-$ is represented by $\sigma_- = \sigma(1 - \lambda)$ and $\theta_- = \theta - \Delta\theta_-$;

(4)    In consistency with the resonant conditions (2), the parameters of the configuration are

$$\lambda = 0.25, \ \Delta\theta_+ = 11.5°, \text{ and } \Delta\theta_- = 33.6°. \tag{5}$$

The nonlinear transfer at all the mentioned **k**-points takes the form [3]

$$NL(\mathbf{k}_-) = I(\mathbf{k}, \mathbf{k}_+, \mathbf{k}_-), \ NL(\mathbf{k}_+) = I(\mathbf{k}, \mathbf{k}_+, \mathbf{k}_-), \ NL(\mathbf{k}) = -2I(\mathbf{k}, \mathbf{k}_+, \mathbf{k}_-), \tag{6}$$

where

$$I(\mathbf{k}, \mathbf{k}_+, \mathbf{k}_-) = Cg^{-8}\sigma^{19}\left[N^2(\mathbf{k})(N(\mathbf{k}_+) + N(\mathbf{k}_-)) - 2N(\mathbf{k})N(\mathbf{k}_+)N(\mathbf{k}_-)\right]. \tag{7}$$

In Equation (6), the fitting dimensional constant, *C*, proportional to the integration **k**-space, has value *C* = 3000 which is valid for the integration grid used by Hasselmann et al., [3]. (The value of C depends not only on the grid resolution parameters but on the kind of the source term of the model, as well). The net nonlinear transfer at any fixed $(\sigma, \theta)$-point of the numerical grid is found by running external vector **k** in Equations (6) and (7) through all the points of the frequency–angle integration grid $\{\sigma_i, \theta_j\}$.

Note that the set of Equation (6) provides the conservative feature for the DIA [3] regardless the location of the quadruplet with respect to the singular subsurface.

The main advantage of this approximation is its evident simplicity. As the analogue of the method of "the integration-in-mean", the DIA has certain accuracy for a certain initial spectrum [3]. The mean error of the DIA (relative to the exact calculations of $I_{\mathrm{NL}}$) is about 60%, if estimated on the ensemble of different wave-spectra shapes [12]. This is the first shortage of the original DIA.

Nevertheless, due to the several orders increase of the speed of calculation for NL-term, the DIA is widely used in practical wind–wave modeling [1]. The third-generation wind–wave numerical models, WAM [13] and WAVEWATCH (WW) [14], are the examples of successful implementation of the DIA.

The other technical shortage of the DIA routine resides in a presence of intermediate and cumbersome interpolation procedures induced by the mismatch of the spectral grid nodes and vectors $\mathbf{k}_+$, $\mathbf{k}_-$ (see Equation (4)). This leads to the time-consuming about 50% CPU for the nonlinear evolution-term calculation during the numerical simulation of wave-field evolution [12].

The radical improvement of the DIA was done by Polnikov and Farina [12], who located all the interacting wave vectors of the DIA configuration exactly at the nodes of the frequency–angular grid, $\{\sigma_i, \theta_j\}$, used in both the kinetic integral and numerical model under application. This provides a

nearly two-times enhancement of the speed of numerical calculation for the *NL*-term in a wind–wave model (with preservation of the accuracy). For this reason, the proposed version of the DIA was called as the fast DIA (FDIA).

Below, we demonstrate all details of the FDIA elements, based on several frequency–angular numerical grids of high-resolution. The aim of this demonstration was to stimulate an active implementation of the FDIA in modern wind–wave models.

## 3. The Concept of the FDIA

In the original version of DIA [3], two of four interacting vectors (i.e., $\mathbf{k}_+$, $\mathbf{k}_-$) are not located at the nodes of integrating grid, what leads to the necessity of the spectrum interpolation. For this reason, the speed of numerical wave forecast calculations is remarkably reduced. The main idea of the FDIA is to use quadruplets which are adjusted to the integration grid for the kinetic integral. To specify this idea, first of all, one should introduce the principal parameters of the grid. Then, the features of configurations in FDIA could be described.

### 3.1. The Integration Grid Properties

Integration grid for kinetic integral will be considered in the polar co-ordinates where each of interacting wave vector $\mathbf{k}_i$ is represented by the frequency–angular point $(\sigma_i, \theta_i)$. Usually [1,3], the integration grid is given by the formulas

$$\sigma_i = \sigma_0 \cdot q^{i-1} \quad (1 \leq i \leq I), \tag{8a}$$

$$\theta_j = -\pi + (j-1) \cdot \Delta\theta \quad (1 \leq j \leq J \text{ and } \Delta\theta = 2\pi/J). \tag{8b}$$

Thus, parameters of the grid are as follows:

- the lowest frequency, $\sigma_0$;
- the frequency exponential increment, $q$;
- the maximum number of frequencies, $I$;
- the angle resolution in radians, $\Delta\theta$;
- and the maximum number of angles, $J$.

To our aims, the principal parameters are $q$ and $\Delta\theta$, as far as they define the resolution of the grid. The numbers $I$ and $J$ should be rather great (several tens), but for the concept under consideration their explicit values $I$ and $J$ are not principal. Note only that the FDIA concept is valid for the rather fine grid (to save an accuracy) when

$$q \leq 1.1 \text{ and } \Delta\theta \leq \pi/10. \tag{9}$$

Everywhere below, the restriction (8) is supposed to be met. Initially, the FDIA was proposed in [12] for the resolution parameters

$$q = 1.05 \text{ and } \Delta\theta = \pi/18, \tag{10}$$

that is called as the "standard" integration grid which was used in [12] for the exact calculation of the kinetic integral $I_{\mathrm{NL}}$ based on the author's method described in [15].

### 3.2. The Choice of Configuration

In the FDIA, the so called "basic configuration", that is the closest to the original DIA, is described by the following ratios. (Pay attention that in the FDIA, a choice of the independent vectors of a quadruplet is changed, accepting $\mathbf{k}_4$ as the external variable).

(1) First, we fix the external vector of integration

$$\mathbf{k}_4 = (\sigma_4, \theta_4), \tag{11a}$$

which is located at a current grid node $(\sigma_4, \theta_4)$ represented in polar co-ordinates.

(2) Then, we fix vector

$$\mathbf{k}_3 = (\sigma_3, \theta_3) \ (\text{with } \theta_3 = \theta_4 + \Delta\theta_{34}), \tag{11b}$$

which is also located at numerical grid node. Here, the new parameter of the configuration, $\Delta\theta_{34} = \theta_3 - \theta_4$, is the angle between vectors $\mathbf{k}_4$ and $\mathbf{k}_3$. These two vectors make the summary vector $\mathbf{k}_a = \mathbf{k}_4 + \mathbf{k}_3$ as a benchmark for directions for the other two vectors, as far as all the vectors of a quadruple are to be allocated in the vicinity of the resonance "figure-of-eight" in the $\mathbf{k}$-space (Figure 1) [3]. In the original DIA, vectors $\mathbf{k}_1$ and $\mathbf{k}_2$ are simply located on this vector $\mathbf{k}_a = (k_a, \theta_a)$ making the basis for the DIA configuration. This is the principle difference between quadruplets in the DIA and the FDIA

(3) Finally, we choose vectors $\mathbf{k}_1$ and $\mathbf{k}_2$, which are also to be allocated at the nodes of the grid, to be directed closely to the direction of vector $\mathbf{k}_a$, what is defined by the ratios

$$\mathbf{k}_1 \approx \mathbf{k}_2 \approx (\mathbf{k}_4 + \mathbf{k}_3)/2 \equiv \mathbf{k}_a/2. \tag{11c}$$

Vector $\mathbf{k}_a$ plays the role of the reference direction along the angle $\theta_a = \theta_4 + \Delta\theta_{a4}$, where parameters $k_a$ and $\sigma_a$, in terms of the independent variables, $\sigma_4$, $\sigma_3$, and $\Delta\theta_{34}$, have the kind

$$\sigma_a = \sigma_4 + \sigma_3. \tag{12}$$

and

$$k_a = \left[\sigma_4^4 + \sigma_3^4 + 2\sigma^2\sigma_3^2\cos(\Delta\theta_{34})\right]^{1/2}. \tag{13}$$

Herewith, the difference between angles $\theta_a$ and $\theta_4$ is given by the ratio

$$\Delta\theta_{a4} = \text{arctg}\left[\frac{\sigma_3^2\sin(\Delta\theta_{34})}{\sigma_3^2\cos(\Delta\theta_{34}) + \sigma_4^2}\right], \tag{14}$$

whilst the correspondence of the quadruplet location near the figure-of-eight is given by the ratio [16]:

$$k_a = \sigma_a^2/2. \tag{15}$$

Thus, after fixing vectors $\mathbf{k}_4$ and $\mathbf{k}_3$, and determining $\Delta\theta_{34}$, Equations (11)–(14) determine the values $k_a$ and $\sigma_a$ for the given $\sigma_4$ and $\sigma_3$. After that, the expression for $\theta_a = \theta_4 + \Delta\theta_{a4}$ finalize possibilities to choose vectors $\mathbf{k}_1$ and $\mathbf{k}_2$. Varying independent parameters for $\mathbf{k}_4$, $\mathbf{k}_3$ and $\Delta\theta_{34}$ (below they are called as "general"), one can vary the values for the dependent parameters $\theta_a$, $k_a$, and $\sigma_a$, determining possible positions for vectors $\mathbf{k}_1$ and $\mathbf{k}_2$.

The main differences between the configurations used in the FDIA and the original DIA are as follows:

(a) All wave vectors $\mathbf{k}_1$, $\mathbf{k}_2$, $\mathbf{k}_3$, and $\mathbf{k}_4$ should be allocated at the nodes of the integration grid;

(b) vectors $\mathbf{k}_1$ and $\mathbf{k}_2$ may be unequal, i.e., they may have some (but small) discrepancies in both values and directions (Figure 2);

(c) the resonance conditions (2) may be rather approximately met, and the quadruplet may be unclosed (Figure 2).

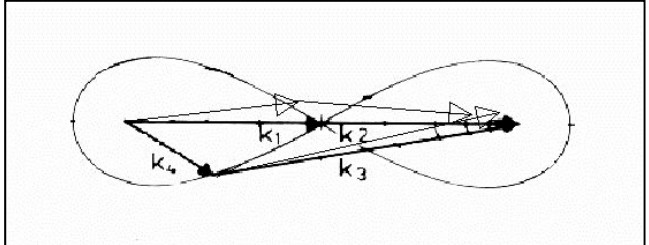

**Figure 2.** Scheme of the fast DIA (FDIA) configuration (thin arrows) on the background of configuration for the original DIA. Vector $\mathbf{k}_4 = (\sigma_4, \theta_4)$ is taken as the external one in the FDIA configuration.

### 3.3. Specification of the Configuration Parameters

To specify the FDIA configuration, it needs to define several integer values corresponding to the abovementioned requirement (a) (allocation of the vectors on the grid). According to the grid (7a), this requirement can be expressed via a set of integer digits by the following equations:

$$\sigma_3 = \sigma_4 \cdot q^{m3}, \ \sigma_1 = \sigma_4 \cdot q^{m1}, \ \sigma_2 = \sigma_4 \cdot q^{m2}, \tag{16a}$$

and

$$\Delta\theta_{34} = n3 \cdot \Delta\theta, \quad \Delta\theta_{a4} = na \cdot \Delta\theta \tag{16b}$$

Here $m1$, $m2$, $n3$, and $na$ are the integer values to be found for any given integer $m3$. The first two are found from requirement (10c), and the latter two do from formulas

$$n3 = \text{Int}(\Delta\theta_{34}/\Delta\theta), \ na = \text{Int}(\Delta\theta_{a4}/\Delta\theta) \tag{16c}$$

via the previously determined values for $\Delta\theta_{34}$ and $\Delta\theta_{a4}$ (as it is described above). In (16c), the function Int( ... ) means the integer number nearest to the value of the argument.

Requirement (b) (inequality of vectors $\mathbf{k}_1$ and $\mathbf{k}_2$) means that one can use the following choice for modulus parameters of the vectors $|\mathbf{k}_1|$ and $|\mathbf{k}_2|$, defined via $\sigma_1$ and $\sigma_2$:

$$m1 = m2 \ \text{or} \ m1 = m2 \pm 1, \tag{17}$$

and the corresponding choice for the angle parameters of the vectors $\mathbf{k}_1$ and $\mathbf{k}_2$:

$$n1 = n2 = na \ \text{or} \ n1 = n2 \pm 1 = na \pm 1 \tag{18}$$

where $n1$, $n2$ are the angular parameters of the vectors $\mathbf{k}_1$ and $\mathbf{k}_2$ corresponding to Equations (11c) and (16b). Sign ($\pm$) means the permutation symmetry for vectors $\mathbf{k}_1$ and $\mathbf{k}_2$.

The choice of ($\pm1$) means a possible inequality of vectors $\mathbf{k}_1$ and $\mathbf{k}_2$. Equations (16b) and (18) mean that for a certain configuration, given by values $m1$, $m2$, $m3$, and $n1$, $n2$, $n3$, the angle parameters of interacting vectors have the values

$$\theta_3 - \theta_4 = \pm n3 \cdot \Delta\theta; \ \theta_1 - \theta_4 = \pm n1 \cdot \Delta\theta; \ \theta_2 - \theta_4 = \pm n2 \cdot \Delta\theta \tag{19}$$

where sing ($\pm$) denotes a set of two mirror configurations due to the matrix $M_{1,2,3,4}$ symmetry (see [3]).

Taking into account the change of the interacting wave vectors order, the net expression for *NL*-term in the FDIA (for the energy-spectrum representation: $\text{NL}_S(\mathbf{k}) \equiv \partial S(\mathbf{k})/\partial t$) is given in the $(\sigma, \theta)$—coordinates by the formulas

$$\text{NL}_S(\sigma_4, \theta_4) = I(\sigma_1, \theta_1, \sigma_2, \theta_2, \sigma_3, \theta_3, \sigma_4, \theta_4), \tag{20a}$$

$$\text{NL}_S(\sigma_3, \theta_3) = I(\sigma_1, \theta_1, \sigma_2, \theta_2, \sigma_3, \theta_3, \sigma_4, \theta_4), \tag{20b}$$

$$\text{NL}_S(\sigma_1, \theta_1) = -I(\sigma_1, \theta_1, \sigma_2, \theta_2, \sigma_3, \theta_3, \sigma_4, \theta_4), \tag{20c}$$

$$\text{NL}_S(\sigma_2, \theta_2) = -I(\sigma_1, \theta_1, \sigma_2, \theta_2, \sigma_3, \theta_3, \sigma_4, \theta_4), \tag{20d}$$

where

$$
\begin{aligned}
I(\sigma_1, \theta_1, \sigma_2, \theta_2, \sigma_3, \theta_3, \sigma_4, \theta_4) = \\
= Cg^{-4}\sigma^{11}\left[S_1 S_2\left(S_3 + (\sigma_3/\sigma_4)^4 S_4\right) - S_3 S_4\left((\sigma_2/\sigma_4)^4 S_1 + (\sigma_1/\sigma_4)^4 S_2\right)\right]
\end{aligned}
\tag{21}
$$

$S_i = S(\sigma_i, \theta_i) = \frac{\sigma_i^4}{4\pi^2 g^3}N(\mathbf{k}_i)$ [12], and the dimensional fitting constant $C$ is depending on the grid parameters. The final 2D-function for NL-term is found by the running vector $(\sigma_4, \theta_4)$ in Equations (20) and (21) through the whole integration grid $\{\sigma_i, \theta_j\}$, similarly to the original DIA procedure.

After some numerical simulations for the grid (7a,b) with parameters $q = 1.1$, $\Delta\theta = \pi/12$ (typical for the WAM), the fitting constant $C$ in (20) is found to be equal to 12,000 [17]. In our case, the change of $C$ is related to the change of the quadruplet configuration (and due to other evolution terms in the wind–wave model used). (This fitting coefficient C is tuned to the total source function of the wind–wave model proposed in [17]).

Hereby, the algorithm of the FDIA configuration calculations is fully described. The certain set of configurations will be given in the next section. It needs only to add that effectiveness of the FDIA against DIA was numerously and successfully verified in comparison with the WAM [17–20] and the WW [21].

## 4. FDIA Parameters for Several Certain Configurations and Grid Resolutions

### 4.1. Parameters of Configuration

On the example of typical WAM integration grid with parameters

$$q = 1.1 \text{ and } \Delta\theta = \pi/12 \text{ (or } \Delta\theta = 15°\text{)}, \tag{22}$$

we shall demonstrate the choice of configuration parameters for FDIA: $m1$, $m2$, $m3$, and $n1$, $n2$, $n3$.

For this aim, from Equations (12)–(14), and (16a), we calculate values $\Delta\theta_{34}$ and $\Delta\theta_{a4}$, varying independently the value of $m3$. Results of calculations are given in Table 1.

**Table 1.** Principle and auxiliary parameters for grid (22). The shaded line has parameters most closely corresponding to the original DIA configuration used in the WAM.

| $m3$ | $\Delta\theta_{34}$, deg. | $\Delta\theta_{a4}$, deg. | $x = \frac{\log(\omega_a/2)}{\log(q)}$ | $m2 = \text{Int}(x + 0.5)$ |
|---|---|---|---|---|
| 3 | 23.8 | 15.2 | 1.61 | 2 |
| 4 | 32.3 | 22.2 | 2.19 | 2 |
| 5 | 41.5 | 30.3 | 2.79 | 3 |
| 6 | 51.6 | 39.8 | 3.42 | 3 |
| 7 | 62.7 | 50.9 | 4.07 | 4 |

From Table 1, it is seen that for the grid (22), the case with $m3 = 5$ (shaded) is the most close to the original DIA configuration for which angle $\Delta\theta_{34} \equiv \theta_- + \theta_+ \cong 45°$ (see Equation (5)).

### 4.2. Parameters for Several Efficient FDIA Configurations

To compare the efficiency of any DIA configurations, a certain criterion was derived in [12], based on accuracy of an approximation (as a measure of discrepancy between approximated and exact estimations of $I_{\text{NL}}$) and the time of calculation for a tested DIA configuration. Owing to this criterion, we found several the most efficient FDIA configurations presented below.

Case 1.

For the original DIA configuration, the following FDIA parameters are the most efficient

$$m3 = 5, m2 = m1 = 3; \tag{23a}$$

and

$$n3 = \text{Int}(\Delta\theta_{34}/\Delta\theta) = 3; \ n2 = n1 = na = \text{Int}(\Delta\theta_{a4}/\Delta\theta) = 2. \tag{23b}$$

Case 2.

If we adopt existence of unequal values $\sigma_1$, $\sigma_2$, then parameters could be

$$m3 = 5, m2 = 2, m1 = 3; \text{ and } n3 = 3, n2 = 3, n1 = 2 \tag{24a}$$

or

$$m3 = 5, m2 = 2, m1 = 3; \text{ and } n3 = 3, n2 = 3, n1 = 1. \tag{24b}$$

In the case of unequal $\mathbf{k}_1$ and $\mathbf{k}_2$, the configuration has a symmetry with respect to permutation $k_1 \leftrightarrow k_2$, that means the possible permutation $(m2, n2) \leftrightarrow (m1, n1)$, making the new quadruplet.

Pay attention that from Table 1 some other configurations are seen which could be used for the DIA. As it was shown in [22], some of them are more efficient than that given by (23a,b) for the original DIA. The relative efficiency for these configurations was checked by means of the special method constructed to this task and presented in [12].

Case 3.

For the multiple, three-configuration DIA (so called the 3C-DIA), the FDIA is the most efficient with the following joint three constituents (with the equal weight for each) [22]

$$(1) \ m3 = 4, m1 = 2, m2 = 3; \text{ and } n3 = 2, n2 = 2, n1 = 1; \tag{25a}$$

$$(2) \ m3 = 5, m1 = m2 = 3; \text{ and } n3 = 3, n2 = n1 = 2; \tag{25b}$$

$$(3) \ m3 = 7, m1 = m2 = 4; \text{ and } n3 = 4, n2 = n1 = 3. \tag{25c}$$

Case 4.

For the grid parameters with more fine angular resolution as

$$q = 1.1 \text{ and } \Delta\theta = \pi/18 \text{ (or } \Delta\theta = 10°), \tag{26}$$

Table 1 has the same kind, whilst the proper configurations are as follows.

For the original DIA, the proper configuration in the FDIA version has parameters

$$m3 = 5, m2 = m1 = 3; \text{ and } n3 = 4, n2 = n1 = 3. \tag{27}$$

The modified FDIA of the most efficiency has the configurations of the type

$$m3 = 5, m2 = 2, m1 = 3; \text{ and } n3 = 4, n2 = n1 = 3 \tag{28}$$

or

$$m3 = 5, m2 = 2, m1 = 3; \text{ and } n3 = 4, n2 = 4, n1 = 3, \tag{29}$$

and some others, corresponding to modifications (18), (19) for unequal $\mathbf{k}_1$ and $\mathbf{k}_2$.

Case 5.

For the multiple 3C-DIA version, the FDIA is the most efficient with the following joint three constituents (with the equal weigh) [22]

$$(1)\ m3 = 4,\ m1 = m2 = 2;\ \text{and } n3 = 3,\ n2 = n1 = 2; \tag{30a}$$

$$(2)\ m3 = 5,\ m1 = m2 = 3;\ \text{and } n3 = 4,\ n2 = n1 = 3; \tag{30b}$$

$$(3)\ m3 = 7,\ m1 = m2 = 4;\ \text{and } n3 = 6,\ n2 = n1 = 5. \tag{30c}$$

## 5. FDIA Parameters of Configurations for a Very High-Resolution Grid

### 5.1. Single Configuration

For applications which can be applied in the future, the following very high-resolution grid is preferable

$$q = 1.05 \text{ and } \Delta\theta = \pi/18 \text{ (or } \Delta\theta = 10°). \tag{31}$$

Principal and auxiliary parameters for this grid are presented in Table 2.

**Table 2.** Principle and auxiliary parameters for grid (31). The shaded line has parameters most closely corresponding to the original DIA configuration used in the WAM.

| $m3$ | $\Delta\theta_{34}$, deg. | $\Delta\theta_{a4}$, deg. | $x = \frac{\log(\omega_a/2)}{\log(q)}$ | $m2 = \text{Int}(x + 0.5)$ |
|---|---|---|---|---|
| 5 | 20.1 | 12.5 | 2.65 | 3 |
| 6 | 24.4 | 15.7 | 3.22 | 3 |
| 7 | 28.7 | 19.2 | 3.80 | 4 |
| 8 | 33.2 | 22.9 | 4.39 | 4 |
| 9 | 37.8 | 27.0 | 4.99 | 5 |
| 10 | 42.7 | 31.4 | 5.60 | 6 |
| 11 | 47.7 | 36.1 | 6.23 | 6 |
| 12 | 53.1 | 41.3 | 6.87 | 7 |
| 13 | 58.7 | 46.9 | 7.51 | 8 |
| 14 | 64.7 | 53.0 | 8.17 | 8 |
| 15 | 71.2 | 59.7 | 8.84 | 9 |

Here within, in the case of the grid (31), the most efficient FDIA single configurations, which could be used in practice, are presented in Table 3 (for a proof of relative efficiency of these configurations among all other configurations, see [22]).

**Table 3.** Principle parameters for the set of the most efficient single configurations for grid (31).

| Index of Configuration | $m3$ (General) | $m1$ | $m2$ | $n3$ (General) | $na$ (General) | $n1$ | $n2$ |
|---|---|---|---|---|---|---|---|
| S1 | 8 | 4 | 5 | 3 | 2 | 2 | 2 |
| S2 | 8 | 4 | 5 | 3 | 2 | 3 | 2 |
| S3 | 9 | 5 | 5 | 4 | 3 | 3 | 3 |
| S4 * | 9 | 4 | 5 | 4 | 3 | 3 | 2 |
| S5 | 10 | 5 | 6 | 4 | 3 | 3 | 3 |
| S6 | 10 | 6 | 6 | 4 | 3 | 3 | 3 |

Notes. 1. Index of configuration includes the symbol of the single configuration type, S, and the conventional number of configuration (for notations, see [14]). Supindex "*" means that the configuration has unequal $n1$ and $n2$; 2. Configuration S6 is marked as the closest one to the original DIA configuration; 3. Parameters $m3$, $n3$, $na$ are marked as "general" as far as $m3$ is independent parameter, and $n3$, $na$ are directly defined by formulas (15a,b) and constant for a given $m3$.

### 5.2. Multiple Constructions of Single Configurations

Finally, we add that some multiple configurations (i.e., constructions of several single configurations [22]) which are more efficient than the simple configurations mentioned in Table 3. These constructions are presented in Tables 4 and 5 given for auxiliary configurations.

**Table 4.** The set of the most efficient double-configuration constructions.

| Index of Construction | Composition of Two Simple Configurations |
|---|---|
| M5 | S1 + S8 |
| M6 | S1 + 0.7·S8 * |
| M7 | S1 + S10 |
| M8 | S1 + 0.7·S10 |

Note. The coefficient in front of configuration means the weight of a proper single configuration from Tables 3 and 5. Supindex "*" means that the configuration has unequal $n1$ and $n2$.

**Table 5.** Auxiliary simple configurations.

| Index of Configuration | $m3$ (General) | $m1$ | $m2$ | $n3$ (General) | $na$ (General) | $n1$ | $n2$ |
|---|---|---|---|---|---|---|---|
| S8 * | 11 | 6 | 7 | 5 | 4 | 4 | 3 |
| S10 | 12 | 7 | 7 | 5 | 4 | 4 | 4 |

Notes. For legend, see notes for Table 3. Super-index "*" means that the configuration has unequal $n1$ and $n2$. These are the parameters of the auxiliary simple configurations used in Table 4.

### 5.3. The 3C-DIA Construction

Finally, it is worthwhile to mention one 3C-DIA construction for the grid (31)

$$(1) \ m3 = 8, \ m1 = 4, \ m2 = 5; \ n3 = 3, \ n1 = n2 = na = 2; \ (\text{config. S1 from Table 2}) \tag{32a}$$

$$(2) \ m3 = 10, \ m1 = 5, \ m2 = 6; \ n3 = 4, \ n1 = n2 = na = 3; \ (\text{config. S5 from Table 2}) \tag{32b}$$

$$(3) \ m3 = 12, \ m1 = 7, \ m2 = 7; \ n3 = 5, \ n1 = n2 = na = 4; \ (\text{config. S10 from Table 5}) \tag{32c}$$

This construction is more effective than one for the original DIA (see Table 6), but it is less effective than ones given in Table 4.

**Table 6.** Efficiency parameters for the constructions considered for the grid (31).

| Index of Construction | S1 | S2 * | S3 | S4 * | S6 (Original DIA) | M5 | M6 | M7 | M8 | 3C-FDIA |
|---|---|---|---|---|---|---|---|---|---|---|
| $Eff_1$ | 5.26 | 6.07 | 4.98 | 5.82 | 4.3 | - | - | - | - | - |
| $Eff_2$ | - | - | - | - | - | 6.57 | 6.43 | 6.43 | 6.39 | 4.4 |

Note. Values of the efficiency parameter $Eff_1$ are applicable for simple configurations, whilst values $Eff_2$ do to two-configuration constructions [12,20]. Supindex "*" means that the configuration has unequal $n1$ and $n2$.

### 5.4. Remarks on the Efficiency

For the completeness of the text, the digitized values of the conventional efficiency parameter, *Eff*, for the abovementioned single configurations and multiple constructions are presented in Table 6.

## 6. Discussion

The DIA was proposed in 1985 [3], and for a long time was unchanged for the reasons of complexity of the point. Some ideas of improving the DIA were declared by van Vledder in [7], but the radical step was made by Polnikov and Farina in [12]. This was possible due to owning the routine for the exact calculation of the kinetic integral [15], that allows formulating the criterion of comparing an efficiency of different versions for DIA and its modifications. Finally, the idea of locating the interacting wave vectors at the nodes of the numerical grid was proposed and realized in [12]. Despite of mismatch of the exact resonance conditions (2), the conservative feature of the NL-term is saved in the FDIA due to ratios (20) (analogous to ratios (6)).

It is found that this modification provides not only an enhancement of the speed of calculation of the NL-term but has better accuracy as well. The calculation speed is increased due to eliminating the interpolation procedures in the original DIA, whilst the better accuracy of FDIA is due to the better choice of the quadruplet configuration [12,22].

This double positive effect in calculation of the NL-term is due the fact of rather crude efficiency of the original DIA (the mean error is about 60% [12]), and better choice of the configuration (see Sections 4.1 and 4.2 above). For the NL-term, the FDIA provides the increase of accuracy in 20%, whilst the speed of calculation is enhanced nearly twice. The tables of comparison for the accuracy and time-consuming values of FDIA and DIA are not given here to save the room of this paper. They are presented in the numerous early papers, both for the net NL-term [12,22] and for the real wind–wave models WAM and WW [18–21].

Based on these results, the FDIA was implemented in the National Institute of Oceanography in India [20]. It is still left to spread this positive result to the modern versions of the world-wide used models: WAM and WW. The present paper aimed to prompt this implementation.

## 7. Conclusions

Details of the original discrete interaction approximation (DIA) are presented, and the concept of the fast DIA (FDIA) is comprehensively described.

Numerous versions of the FDIA configurations for different numerical grids are presented, including the single and multiple configurations in a high-resolution case.

The preference of the FDIA compared to the original DIA in accuracy and time-consuming are mentioned and explained. Some estimations of increased efficiency of the FDIA are shown.

**Funding:** This research received no external funding.

**Acknowledgments:** I am thankful to the Guest Editor Alberto Alberello for his kind invitation to submit my paper in this special issue and to anonymous reviewers whose comments allow improving the text.

**Conflicts of Interest:** The author declares no conflict of interest.

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
