# Peer review of "The Fast Discrete Interaction Approximation Concept"

_fluids, doi:10.3390/fluids5040176_

Round 1

Reviewer 1 Report

The article is an extension of the author's concepts to date published in major journals and cited many times. It is written logically and consistently and does not raise any objections in terms of the presentation of the substantive concept. There were no errors in the mathematical structure of the article. However, in the opinion of the reviewer, the article requires the following additions:

1. The introduction is very brief, it does not contain a proper introduction to the subject. The author should first discuss the origins of this methodology and its applications. He should also cite the current literature on this subject with a brief overview and demonstration of common features and differences.
2. The literature review concerns mainly historical and author own publications. It should be extended to include current literature from other research centers.
3. It would be very helpful to suplement the article with a complete list of symbols with a detailed description of both the variable name and its meaning and influence on the algorithm.

After these changes, the article will be a valuable scientific contribution of the author to world literature and should be published.

Author Response

Reply to the Reviewer 1.

Remarks 1 and 2, related to the text volume,

are executed by adding 5 new references and a short discussion of them.

Remark 3, related to a list of symbols,

is not executed because it dose not correspond to the template.

Besides, the list of symbols and its meaning could take too much room of the paper.

Reviewer 2 Report

General comments

The topic addressed by the study is interesting and very actual and should be published. Accuracy of the DIA approximation has been a subject of increasing debate in the literature and international forums in the last years. Although the manuscript is consistent as it stands, I miss references to the recent investigations.

Tamura et al. (2010) studied the impact of non-linear energy transfer on realistic wave fields of the Pacific Ocean using the Simplified Research Institute of Applied Mechanics (SRIAM) model, which was developed to accurately reproduce non-linear source terms with lower computational cost than more rigorous algorithms, and the widely used Discrete Interaction Approximation (DIA) method. Comparison of the model with buoy observations revealed a negligible difference in significant wave heights but pronounced bias in peak frequency with DIA. The analysis of spectral shape indicated that the SRIAM method can quantitatively capture the overshoot phenomena around the spectral peak during wave growth.

Tamura H., Waseda T, Miyazawa Y. (2010). Impact of nonlinear energy transfer on the wave field in Pacific hindcast experiments. J. Geophys. Res., Vol. 115, C12036, 20 pp. doi:10.1029/2009JC006014

 Tolman (2013) presented a Generalized Multiple DIA (GMD) which expands upon the DIA by (i) expanding the definition of the representative quadruplet, (ii) formulating the DIA for arbitrary water depths, (iii) providing complimentary deep and shallow water scaling terms and (iv) allowing for multiple representative quadruplets. The GMD is rigorously derived to be an extension of the DIA, and is backward compatible with it. The free parameters of the GMD have been optimized holistically, by optimizing full model behaviour in the WAVEWATCH III wave model (Tolman and Grumbine, 2013). The results showed that in deep water, GMD configurations can be found which remove most of the errors of the DIA. Most of these improvements were implemented in a new version (4.18) of the code.

The two-scale approximation (TSA) to the full Boltzman integral representation of quadruplet wave-wave interactions has recently been presented as a new method to estimate nonlinear transfer rates in wind waves, and has been tested for idealized spectral data, as well as for observed field measurements (Willis and Bonnefond, 2013). TSA has been implemented in WAVEWATCH III and shown to perform well for wave spectra from field measurements, even for cases with directional energy shearing, compared to the DIA.

WaveWatch III® that was released in March 2014 by NOAA/NCEP allowing the use of unstructured grids and introducing new parameterizations for wave dissipation together with new parameterizations for bottom friction including movable bed roughness.

Tolman, H. L. 2013. A Generalized Multiple Discrete Interaction Approximation for resonant four-wave interactions in wind wave models. Ocean Modelling 70, 11-24.

Tolman, H. L. & Grumbine, R. W. 2013. Holistic genetic optimization of a Generalized Multiple Discrete Interaction Approximation for wind waves. Ocean Modelling 70, 25-37.

Tolman, H. L. & Group, T. W. I. D. 2014. User manual and system documentation of WAVEWATCH III® version 4.18.

Willis, J. & Bonnefond, P. 2013. Report of the Ocean Surface Topography Science Team Meeting. In: Boulder, C. (ed.). USA.

The nonlinear source term was discussed at the 12th International Workshop on Wave Hindcasting and Forecasting held in Hawai (2011), and also recently at the 16th International Waves Workshop 10–15 November 2019, Melbourne, Australia.

  Specific comments

  1. I would suggest to write full name of DIA within capital letters: the Discrete Interaction Approximation (DIA).

Author Response

1. Reply to the remark about paper by Tamura (2010).

This work is good, as well as numerous similar to it. Though in our consideration of the FDIA , we do not touch comparison with any measurements in the field, because the field wave evolution is reasonably governed by the input and dissipation terms that provide their impact on wave parameters. The proper our papers, where the FDIA is used, are cited in the references list.

2. Reply to the remark about Tolman (2013). This paper gives a reasonable advance in the DIA elaboration. It is additionally mentioned in new references list, together with reference to van Vledder (2001). 

Nevertheless, detailed discussion of them are not given, as far as they both retained the main shortage of the DIA, i.e. the location of quadruplets at the resonance subsurface, which is excluded in the FDIA . Besides, our optimization of DIA is based on the especially derived criterion. This optimization could enforce the new efforts in elaboration wind-wave models, but, unfortunately, FDIA was never mentioned in numerous researches and papers, since 2003 till present, to say nothing of implementation. Our paper is namely aimed to fill this gap.

3. Reply to the remark about the Two-Scale Approximation (TSA). Several papers about the TSA are included in the list of references. Though these papers are out of improving the DIA, thus they are out of our discussion.

4. Reply to the additional list of references. The author is thankful to the Reviewer for this list. A part of it was included in the revised list of my references.

5. Reply to the specific comments. Recommendation is done.
